# Sustainable Valorization of Industrial Cherry Pomace: A Novel Cascade Approach Using Pulsed Electric Fields and Ultrasound Assisted-Extraction

**DOI:** 10.3390/foods13071043

**Published:** 2024-03-28

**Authors:** Ervehe Rrucaj, Serena Carpentieri, Mariarosa Scognamiglio, Francesco Siano, Giovanna Ferrari, Gianpiero Pataro

**Affiliations:** 1Department of Industrial Engineering, University of Salerno, Via Giovanni Paolo II, 132, 84084 Fisciano, SA, Italyscarpentieri@unisa.it (S.C.); mrscogna@unisa.it (M.S.); gferrari@unisa.it (G.F.); 2ProdAl Scarl, University of Salerno, Via Giovanni Paolo II, 132, 84084 Fisciano, SA, Italy; 3Institute of Food Science, National Research Council (CNR), Via Roma 64, 83100 Avellino, AV, Italy; francesco.siano@isa.cnr.it

**Keywords:** cherry pomace, pulsed electric fields, ultrasounds, cascade approach, phenolic compounds

## Abstract

In this study, a two-stage cascade extraction process utilizing pulsed electric fields (PEF) (3 kV/cm, 10 kJ/kg) for initial extraction, followed by ultrasound (US) (200 W, 20 min)-assisted extraction (UAE) in a 50% (*v*/*v*) ethanol-water mixture (T = 50 °C, t = 60 min), was designed for the efficient release of valuable intracellular compounds from industrial cherry pomace. The extracted compounds were evaluated for total phenolic content (TPC), flavonoid content (FC), total anthocyanin content (TAC), and antioxidant activity (FRAP), and were compared with conventional solid-liquid extraction (SLE). Results showed that the highest release of bioactive compounds occurred in the first stage, which was attributed to the impact of PEF pre-treatment, resulting in significant increases in TPC (79%), FC (79%), TAC (83%), and FRAP values (80%) of the total content observed in the post-cascade PEF-UAE process. The integration of UAE into the cascade process further augmented the extraction efficiency, yielding 21%, 49%, 56%, and 26% increases for TPC, FC, TAC, and FRAP, respectively, as compared to extracts obtained through a second-stage conventional SLE. HPLC analysis identified neochlorogenic acid, 4-p-coumaroylquinic, and cyanidin-3-*O*-rutinoside as the predominant phenolic compounds in both untreated and cascade-treated cherry pomace extracts, and no degradation of the specific compounds occurred upon PEF and US application. SEM analysis revealed microstructural changes in cherry pomace induced by PEF and UAE treatments, enhancing the porosity and facilitating the extraction process. The study suggests the efficiency of the proposed cascade PEF-UAE extraction approach for phenolic compounds from industrial cherry pomace with potential applications to other plant-based biomasses.

## 1. Introduction

The sweet cherry (*Prunus avium* L.) is esteemed worldwide for its delicious flavor, nutritional abundance, and health-enhancing properties. Cultivated extensively, particularly in the Mediterranean region and notably in Apulia, Italy, the ‘Ferrovia’ cultivar produces large, heart-shaped fruits with vibrant red skin, firm flesh, and juicy pink pulp tightly embracing the stone [1]. While predominantly enjoyed fresh, about 15% of the European Union’s cherry production undergoes processing, notably in France, Italy, and Spain. Processed forms include canned cherries in syrup, alcoholic confectionery preparations, sugar-based delights like candied fruit and jams [2,3], and fruit juices that have been increasingly favored for their reported nutritional and health benefits [4,5,6,7,8,9].

However, industrial cherry processing inevitably generates substantial waste, including seeds, stems, and pomace—primarily composed of skin and remaining flesh. Cherry pomace, or press cake, accounts for 15–28% of the initial fruit weight [10] and is typically produced during mechanical pressing operations [11]. Rich in lignin, cellulose, dietary fiber, and phenolic compounds such as neochlorogenic acid, chlorogenic acid, 4-p-coumaroylquinic acid, and rutin, cherry pomace also serves as a valuable source of anthocyanins like cyanidin-3-*O*-rutinoside and cyanidin-3-*O*-glucoside [11,12]. These compounds contribute to various biological activities, including antioxidant, anti-inflammatory, anticancer, α-glucosidase inhibiting, and antihypertensive properties [13,14,15,16].

Given the increasing availability of cherry pomace and its diverse array of beneficial compounds, this plant biomass holds promise for valorization through cascade processing. This approach involves a multi-stage extraction process designed to efficiently recover a spectrum of valuable compounds while adhering to sustainable practices aimed at minimizing environmental impact and energy consumption [17].

Traditional extraction methods such as percolation, maceration, and Soxhlet extraction often necessitate cell disintegration pre-treatment to enhance the release of target intracellular compounds efficiently at low temperatures, thereby reducing energy costs, solvent usage, and extraction duration [18,19]. Pulsed electric fields (PEF) and ultrasound (US) pre-treatments have emerged as promising techniques for gentle and more effective cell disruption in plant-based biomass, respectively, intensifying the release of bioactive compounds from various agro-food waste and byproducts during subsequent solid-liquid extraction (SLE) [20,21,22].

PEF pre-treatment entails exposing moist plant tissues to brief, repetitive pulses of moderate electric field intensity (ranging from 0.5 to 10 kV/cm) and low-energy input (from 1 to 20 kJ/kg), inducing cell membrane permeabilization through electroporation or electropermeabilization [23]. This facilitates the transfer of intracellular compounds into the extracting medium, particularly those of small molecular weight or those not tightly bound to the intracellular structure. However, extraction of higher molecular weight molecules or those more tightly bound may necessitate additional techniques such as ultrasound or high-pressure homogenization [24].

Ultrasound treatment, either alone or assisting in SLE, utilizes the acoustic cavitation effects produced in the treatment medium by the passage of US waves ranging from 18 to 200 kHz [25] enhance surface contact between solvents and plant tissue, accelerating the diffusion of solutes into the solvent [22,26]. When implemented under optimal conditions and with measures taken to prevent excessive heating of the treated sample, this technique ensures efficient extraction without compromising the quality of the extracts [22].

Several studies have demonstrated the effectiveness of PEF-assisted extraction and ultrasound-assisted extraction (UAE), applied individually, in recovering natural pigments and phenolic compounds from various food and food byproducts [22,27,28,29,30,31,32], including cherry fruit and pomace [12,28,33,34,35,36], while reducing energy consumption and extraction time [22,32,37]. Moreover, the combination of PEF and UAE has shown enhanced extraction efficiency of phenolic compounds from different matrices, such as fresh rosemary and thyme byproducts [38] almonds [39], and grapefruit [40]. However, as per the literature survey, no study has been published yet on the use of PEF and US in a cascade approach for maximizing the recovery of high-value compounds from cherry pomace. Moreover, it has been shown that the combination of PEF-assisted extraction and UAE could lead to the enhanced extraction efficiency of phenolic compounds recovered from different matrices.

Integrating PEF and US treatments for cherry pomace valorization may offer a promising approach for exploiting PEF’s selectivity for recovering low molecular weight and less-bounded intracellular compounds and for utilizing US’s effectiveness in disrupting cell walls and membranes, thus enabling the recovery of bioactive compounds remaining as complexes in residual biomass. Additionally, using these technologies in a cascade approach could allow milder processing conditions compared to individual applications, minimizing process severity and energy requirements.

This study aims to investigate the sequential and cumulative effects of PEF and US-assisted extraction in a cascade approach, enhancing cellular disruption and intensifying the recovery of total phenolic compounds, flavonoids, and anthocyanins from industrial cherry pomace during ethanol-water diffusion. Specifically, the impact of individual PEF and cascaded (PEF+US) treatments on cherry pomace morphology via SEM analysis and on phenolic compound extraction yield and composition via HPLC-DAD analysis was assessed.

## 2. Materials and Methods

### 2.1. Raw Materials and Chemicals

Industrial cherry pomace (press cake) of the “Ferrovia” variety was generously provided by a local juice producer during the cherry processing season of 2023. These samples were swiftly transported to the laboratories of ProdAl Scarl (Fisciano, Italy) and stored under refrigerated conditions (T = 4 °C) until use. Upon arrival at the laboratory, the moisture content of the cherry pomace was assessed to be 81.5% on a wet basis.

Ethanol, all of the reagents, and the standards were purchased from Sigma Aldrich (Steinheim, Germany).

### 2.2. PEF and US Equipment

The pre-treatment of industrial cherry pomace with PEF was conducted using a laboratory-scale batch system, as previously described elsewhere [32]. The system comprised a high-voltage pulsed power generator (Modulator PG, ScandiNova, Uppsala, Sweden) capable of delivering monopolar square wave pulses with adjustable pulse widths (ranging from 3 to 25 μs) and frequencies (from 1 to 450 Hz) to the plant tissue within a batch treatment chamber. This chamber consisted of two parallel-plane stainless steel electrodes separated by a Teflon spacer, with a gap of 2 cm and an area of 75 cm². The monitoring of the actual voltage applied across then electrodes and current passing through the treatment chamber was performed using a high-voltage probe (Tektronix, P6015A, Wilsonville, OR, USA) and a Rogowsky coil (2-0.1, Stangenes, Inc., Palo Alto, CA, USA), respectively, connected to a 300 MHz oscilloscope (Tektronix, TDS 3034B, Wilsonville, OR, USA). The maximum electric field intensity (E, in kV/cm) and the total specific energy input (W_T_, in kJ/kg of plant tissues) were determined following the method outlined by Carpentieri, et al. [22].

The US treatments of cherry pomace, post-PEF-assisted extraction, were conducted using a US processor UP 400S (Hielscher GmbH, Chamerau, Germany). This equipment can deliver a maximum power (P) of 400 W by adjusting the amplitude of the oscillatory system between 0 and 100%, while maintaining a constant frequency (f) of 24 kHz. The sonication probe employed was Tip H3 (titanium, 3 mm in diameter), characterized by an acoustic power density of 460 W cm^−2^ [22].

### 2.3. Cascade of Pulsed Electric Fields (PEF)- and Ultrasound (US)-Assisted Extraction Experiments

Figure 1 illustrates the two-step experimental procedure employed for cherry pomace valorization, incorporating the cascaded combination of PEF- and US-assisted extraction (PEF+UAE).

In the first step, approximately 80 g of industrial fresh cherry pomace underwent PEF pre-treatment under established optimal conditions (*E* = 3 kV/cm, *W_T_* = 10 kJ/kg) as defined by Rrucaj et al. [12]. Following the electropermeabilization treatment, the cherry pomace underwent solid-liquid extraction (SLE) according to the protocol outlined in previous studies [12,34]. The combination of PEF pre-treatment and SLE will be referred to as PEF-assisted extraction in the following. Specifically, the PEF-treated cherry pomace was promptly transferred into a 500 mL Erlenmeyer flask containing a 50% (*v*/*v*) ethanol-water mixture as the extracting solvent at a constant solid-to-liquid ratio (1:5 g/mL). The flask was then placed in an orbital incubator S150 (PBI International, Milan, Italy) at 50 °C with constant shaking at 160 rpm for varying diffusion times (5–60 min). Following the incubation, samples were centrifuged at 5700× *g* for 10 min (PK121R model, ALC International, Cologno Monzese, Milan, Italy) to separate the clear supernatant, denoted as the first output stream (labeled as Extract_(I)_ in Figure 1), from the residual cherry pomace.

For comparison, a conventional solid-liquid extraction (SLE_(I)_) process was conducted using the same raw material and extraction protocol but without the PEF pre-treatment.

The residual biomass remaining after the PEF-assisted extraction underwent a second step involving UAE to further enhance the recovery yield of bioactive compounds. The sample was transferred to a 1000 mL Erlenmeyer flask and resuspended in a 50% (*v*/*v*) ethanol-water mixture at a solid-to-liquid ratio of 1/5 g/mL. To prevent excessive heating during the US treatment, the flask was placed in an ice-water bath. Sonication treatments were conducted under optimal conditions (*P* = 200 W, *t* = 20 min, *f* = 24 kHz) as determined by preliminary experiments depicted in Appendix A. The experiments were carried out with the sample at the initial temperature set at 20 ± 1. Throughout the experiments, the maximum temperature rise remained below 5 °C, and the temperature was monitored using a K-type thermocouple inserted into the flask containing the solvent-cherry pomace mixture. After US treatment, the flask was transferred to the incubator set at 50 °C, and the diffusion process continued with constant shaking at 160 rpm for varying diffusion times (5–60 min). Upon completion of the UAE process, the same post-processing steps as the PEF-assisted extraction were conducted to obtain the second output stream (labeled as Extract_(II)_ in Figure 1), separated from the spent residual cherry pomace.

For comparison, a sample of residual biomass from the first PEF-assisted extraction underwent a second conventional solid-liquid extraction (SLE_(II)_) using the same protocol as for UAE but without activating the US processor.

To investigate the effect of the extraction time on the release of phenolic compounds through the two cascaded steps, 2 mL aliquots of extracts were withdrawn from the flasks containing untreated (SLE_(I)_) and PEF-treated cherry pomace (Step I), as well as PEF combined with a second SLE stage (PEF-SLE_(II)_) and US (PEF-UAE). These were collected at various diffusion times (5, 10, 15, 30, 45, and 60 min) and promptly centrifuged at 15,600× *g* using the 5417R Centrifuge (Eppendorf, Hamburg, Germany) for 10 min at 5 °C to separate the supernatants. The final extracts were then stored at 4 °C until further analysis.

### 2.4. Chemical Characterization of the Sweet Cherry Pomace Extracts 

#### 2.4.1. Determination of Total Phenolic Content (TPC)

The TPC of the obtained extracts was determined through the Folin–Ciocalteau assay, following the protocol outlined by Carpentieri et al. [32]. The absorbance of the reacting mixtures containing the diluted samples (with a dilution factor of four) was measured at 765 nm using a UV/Vis spectrophotometer (V-650, Jasco Inc., Easton, MD, USA). A five-point external standard calibration curve was generated using gallic acid dissolved in a 50% (*v*/*v*) ethanol-water mixture, covering a concentration range of 10 to 100 mg/L. Results were expressed in mg of gallic acid equivalent per g of dry matter of cherry pomace (mg GAE/gDW). 

#### 2.4.2. Determination of Flavonoid Content (FC)

The determination of FC of the obtained extracts was performed using the aluminium-chloride colorimetric assay according to the method outlined by Carpentieri et al. [32]. The samples, diluted fourfold, were mixed with reagents, and their absorbance was measured at 510 nm using a V-650 spectrophotometer. Quercetin dissolved in 50% (*v*/*v*) ethanol-water mixture was used to generate a five-point external standard calibration curve covering a concentration range of 20 to 100 mg/L. The results were expressed as mg of quercetin equivalent per g of dry matter of cherry pomace (mg QE/gDW). 

#### 2.4.3. Determination of Total Anthocyanin Content (TAC)

The TAC of the extracts was quantified using the pH differential method according to the method described by Carpentieri et al. [30]. Briefly, each extract was used to prepare two mixtures. The first mixture consisted of the diluted samples (dilution factor of 10) and the pH 1.0 buffer (0.19% *w*/*v* potassium chloride in water). The second mixture consisted of the same diluted sample and the pH 4.5 buffer (5.44% *w*/*v* sodium acetate in water). The absorbance of both mixtures was recorded at wavelengths of 520 nm and 700 nm. The concentration of the anthocyanins, expressed as mg of C3G (cyanidin-3-*O*-glucoside) equivalent per g of dry matter of cherry pomace (mg C3G/g_DW_) was determined by using Equation (1):(1)$$C=\frac{A\times MW\times DF\times 10^{3}}{\varepsilon \times DW}\times \frac{L}{S}$$
where:$$A={\left(A_{520\mathrm{nm}}-A_{700\mathrm{nm}}\right)}_{pH=1}-{\left(A_{520\mathrm{nm}}-A_{700\mathrm{nm}}\right)}_{pH=4.5}$$

MW = molecular weights of cyanidin-3-*O*-glucoside;DF= dilution factor; ε = molar extinction coefficient;DW = dry weight of cherry pomace;10^3^ = conversion factor from g to mg; andL/S = liquid-to-solid ratio.

#### 2.4.4. Determination of Ferric-Reducing Antioxidant Power (FRAP) 

The FRAP assay of cherry pomace extracts was performed following the methodology reported by Carpentieri et al. [32]. The absorbance of the resulting mixtures containing the diluted samples (dilution factor of 10) was recorded at a wavelength of 593 nm. A 50% (*v*/*v*) ethanol-water mixture was used to prepare the standard solutions of ascorbic acid (0 and 2 mmol/L). The antioxidant capacity was expressed as mg of ascorbic acid equivalent per g of dry matter of cherry pomace (mg AAE/g_DW_).

#### 2.4.5. HPLC-DAD Analyses of the Extracts

The compositions of the phenolic compounds present in extracts from untreated (SLE_(I)_), PEF-treated, and cascade-treated samples (PEF-SLE_(II)_, PEF-UAE) were assessed following the methodology outlined by Magri et al. [41]. Briefly, the extracts were separated using an Agilent HP 1100 modular chromatographer (Agilent Technologies, Paolo Alto, CA, USA) equipped with a Jupiter C18 reverse-phase column with 250 × 2.1 mm inner diameter and a particle diameter of 4 mm (Phenomenex, Torrance, CA, USA). The column was maintained at a constant temperature of 37 °C. Separations were carried out at a constant flow rate of 0.2 mL/min, using solvent B (acetonitrile/0.1% trifluoroacetic acid). The following gradient was used: 0–4 min: 0% B; 4–14 min: 0–14% B; 14–30 min: 14–28% B; 30–34 min: 28% B; 34–42 min: 28–60% B; 42–45 min: 60–80% B; 45–50 min: 80–100% B. Solvent A consisted of 0.1% trifluoroacetic acid in HPLC-grade water. The injection volume of the extract for each analysis was 10 μL. 

High-performance liquid chromatography (HPLC) separations were monitored by recording UV-visible spectra every 2 s using a diode array detector (DAD). Chromatograms were acquired at fixed wavelengths (λ) of 520, 360, 320, and 280 nm corresponding to the maximum absorbance of each phenolic compound. ChemStation software version A.10 (Agilent Technologies, CA, USA) was used to process the obtained data. Phenolic compounds in the extracts were identified by comparing the retention times with those of pure standards. Phenolic compounds were quantified by generating calibration curves from standard solutions (R^2^ > 0.99) prepared at six different concentrations (5–250 mg/kg) in methanol and subsequently diluted (10 times) with 0.1% trifluoroacetic acid before injection. The resulting concentrations were expressed as mg of the target compound per g dry matter of cherry pomace.

### 2.5. Scanning Electron Microscopy (SEM) Analysis

The effect of the PEF pretreatment and the cascade process on the morphological characteristics of cherry pomace cell tissues was assessed using Scanning Electron Microscopy (SEM), as described by Pirozzi et al. [42]. The dried cherry pomace was mounted on an aluminum stub and coated by a 10 nm thick gold-palladium alloy sputter coater before being analyzed in a high-resolution ZEISS HD15 Scanning Electron Microscope (Zeiss, Oberkochen, Germany) at 1000× magnification.

### 2.6. Statistical Analysis

All experiments and analyses were performed in triplicate, and the results were reported as means ± standard deviations. To determine statistically significant differences (*p* < 0.05), one-way analysis of variance (ANOVA) and Tukey’s test were carried out, employing the SPSS 20 statistical package (IBM, Chicago, IL, USA). Pearson correlation analysis was employed to assess the strength of the linear correlations between dependent variables, yielding correlation coefficients (r).

## 3. Results and Discussion

### 3.1. PEF-Assisted Extraction Process of Phenolic Compounds from Industrial Cherry Pomace

Figure 2 depicts the influence of the diffusion time on the level of TPC, FC, TAC, and FRAP values of the extracts from untreated and PEF-treated cherry pomace during solvent extraction in 50% (*v*/*v*) ethanol-water mixtures at 50 °C for up to 60 min.

The results from Figure 2a–c indicate that the profiles of the extracts from untreated and PEF-pre-treated samples were similar, regardless of the type of phenolic compounds. Both showed a significant dependence of TPC, FC, and TAC on extraction time, in line with previous findings [12,30]. Initially, the phenolic compound content increased rapidly during conventional extraction (SLE_(I)_), driven by a high concentration gradient between the solid and liquid phases. Additionally, the influence of ethanol and moderate temperature on cell membrane barrier properties likely contributed to improved compound solubility and diffusivity, thereby enhancing extractability [22,30,43,44]. However, extraction in a hydroalcoholic solvent was relatively slow, with limited yield even after prolonged diffusion times. Furthermore, extraction rates decreased with prolonged extraction due to diminishing driving forces and analyte concentration in the solid phase, eventually reaching equilibrium. Maximum phenolic compound content was detected after approximately 60 min of extraction for TPC (41.6 ± 0.02 mgGAE/gDW) and 45 min for both FC (48.6 ± 0.24 mgGAE/gDW) and TAC (0.59 ± 0.01 mgGAE/gDW). This trend aligns with previous studies that indicate a slight decrease in FC and TAC values with prolonged extraction time, which is likely due to oxidation reactions or degradation phenomena accelerated at higher temperatures [30,32,45].

The application of PEF pre-treatment significantly (*p* < 0.05) enhanced phenolic compound recovery by 292–412% for TPC, 215–441% for FC, and 303–471% for TAC as compared to the control extraction (SLE_(I)_) (Figure 2a–c). Extraction times maximizing phenolic compound recovery from PEF-treated cherry pomace were approximately 30 min for TPC, 45 min for FC, and 60 min for TAC, yielding levels of 166.6 ± 0.14 mgGAE/gDW, 152.9 ± 0.91 mgQE/gDW, and 2.6 ±0.07 mgC3G/gDW, respectively. These levels were approximately 4, 3.1, and 4.4 times higher than the highest values observed for the control extraction.

The significant (*p* < 0.05) difference was visually confirmed by photos of extracts (Figure 3), showing color evolution over time for untreated and PEF-treated cherry pomace. Extract color intensity correlates with phenolic compound content. Additionally, in the same figure, the photos of the fresh and residual cherry pomace left after both SLE_(I)_ and PEF-assisted extraction are reported. As it can be inferred, the latter show a lighter brown color as compared with SLE_(I)_ and fresh cherry pomace, which may be attributed to the higher release of phenolic compounds from the PEF-treated samples.

This substantial increase in phenolic compound extraction can be ascribed to the induction of cell membrane electroporation by the PEF treatment, facilitating solvent penetration into the plant cell cytoplasm and the subsequent mass transfer of solubilized intracellular compounds, intensifying phenolic compound extractability [30,46,47].

Although making direct comparisons between phenolic compound extraction yields from the fruit processing residues in previously published papers is challenging due to various factors (e.g., fruit variety, ripening conditions, processing methods of fruits, type of fruit residue, equipment, and experimental extraction protocols), the results from this study align with the literature findings on cherry or other red fruit residues. For example, Rrucaj et al. [12] found that the PEF (3 kV/cm, 10 kJ/kg) pre-treatment of cherry press cake achieved after juice expression from sweet cherry fruits in a laboratory scale press led to significant increases in TPC (+26%), FC (+27%), and TAC (+42%) as compared to conventional SLE. Likewise, Pataro et al. [33] and Bobinaite et al. [34] observed that PEF-induced permeabilization of the cell membranes significantly increased the extraction yield of TAC (38–54%) from sweet and sour cherry press cakes in comparison with the control SLE. However, in these studies, PEF treatment was applied to fresh fruit before the juice expression rather than the resulting press cake.

Similar trends were noted for other fruit residues, suggesting the efficacy of PEF in augmenting phenolic compound extraction [30,32,48].

Furthermore, results from the FRAP assay (Figure 2d), reflecting antioxidant activity, showed a significant increase (*p* ≤ 0.05) in FRAP values with increasing diffusion times, consistent with TPC, FC, and TAC trends. Extracts from PEF-treated samples exhibited significantly higher antioxidant activity (+273–441%) as compared to untreated samples (SLE_(I)_), which was attributed to the high phenolic compound content recovered post-electropermeabilization treatment. The maximum FRAP values were detected after 60 min for untreated samples (34.2 ± 0.03 mmolAAE/100 g FW) and 45 min for PEF-treated samples (129.1 ± 0.09 mmolAAE/100 g FW).

These findings align with those reported by Rrucaj et al. [12], Pataro et al. [33], and Bobinaite et al. [34], all of whom observed heightened antioxidant activity in extracts from PEF-treated cherry press cake compared to untreated samples, showing increases of +44%, +21%, and +24%, respectively. 

Furthermore, a robust positive correlation was noted between the TPC, FC, TAC, and FRAP values, with Pearson correlation coefficients ranging from 0.94 to 1.00 for TPC, 0.79 to 0.99 for FC, and 0.94 to 0.97 for TAC. This indicates that phenolic compounds are the main contributors to the overall antioxidant activity of cherry pomace extracts, consistent with findings from prior research [12,33,49].

The attained results highlight the promising potential of PEF technology as a highly efficient extraction method, enhancing phenolic compound extractability without the need for harmful organic solvents. This innovative approach may establish a foundation for a cascading strategy, enabling subsequent stages for capitalization on enhancements initiated by initial PEF pre-treatment.

Based on these results, subsequent experiments were conducted to explore the impact of the cascade process (PEF+UAE) on the retrieval of valuable compounds from cherry pomace. The PEF-assisted extraction process was set at the optimal extraction time (t = 60 min), determined as the point maximizing all the investigated responses (Figure 2), and designated as the initial step in the cascade before UAE.

### 3.2. Cascade PEF-UAE Extraction Process of Phenolic Compounds from Cherry Pomace

The cascade mode application of PEF and US involved a dual-stage solvent extraction process, aimed at progressively enhancing cell disruption to intensify the extraction yield of phenolic compounds and ensure the recovery of a wider range of valuable components from industrial cherry pomace. 

Figure 4 illustrates the cumulative TPC (Figure 4a), FC (Figure 4b), TAC (Figure 4c), and FRAP values (Figure 4d) of extracts obtained after the cascade application of PEF-assisted 60 min extraction from industrial cherry pomace (Step I), followed by a subsequent extraction step (Step II) using US (*P* = 200 W, *t* = 20 min)-assisted extraction (UAE) of residual cherry pomace in a 50% (*v*/*v*) ethanol-water mixture at 50 °C, as a function of extraction time. For comparison, a sample of residual cherry pomace obtained after the PEF-assisted extraction process underwent a second conventional solid–liquid extraction (SLE_(II)_) step using the same protocol as for UAE but with the US processor switched off.

The cascade combination of PEF and SLE_(II)_ significantly (*p* ≤ 0.05) increased the extractions of TPC, FC, and TAC with increasing diffusion times, with no significant (*p* < 0.05) differences detected only when the diffusion time was increased from 10 to 20 min for both FC and TAC. Specifically, at each diffusion time, the additional SLE_(II)_ step led to further significant (*p* ≤ 0.05) increments in the recovery yields of phenolic compounds by 12.4–17.9% for TPC, 9.7–15.2% for FC, and 6.7–12.0% for TAC, as compared with the total extraction. This additional release was likely promoted by the high concentration gradient between the residual solid and fresh hydroalcoholic solvent added in the second extraction step. The maximum recovery yield was detected after 60 min of extraction for TPC (35.59 ± 0.02 mgGAE/gDW), FC (27.09 ± 0.15 mgGAE/gDW), and TAC (0.352 ± 0.001 mgGAE/gDW).

Similarly, the combination of PEF and US technologies in a cascade scheme resulted in a significant (*p* < 0.05) improvement in the extractability of phenolic compounds from residual cherry pomace with increasing extraction times. No significant (*p* > 0.05) differences were detected only when the diffusion time was extended from 10 to 15 min for TPC. Furthermore, the additional cell disintegration induced by the US treatment of residual cherry pomace positively influenced the extraction yields of phenolic compounds, which were significantly (*p* ≤ 0.05) higher as compared to the SLE_(II)_ samples, regardless of the diffusion time (Figure 4a–c). Specifically, an increase of 21% for TPC, 49% for FC, and 56% for TAC compared to the extracts obtained during SLE_(II)_ were detected in the UAE samples after 60 min of extraction (Figure 4a–c).

Correspondingly, the supernatant from the UAE samples showed a slightly more intense light red color than the one obtained from SLE_(II)_, corroborating the ability of US pretreatment to further unlock the phenolic compounds strongly bound or likely remaining trapped inside the cell during the first extraction step. This is also supported by the slightly lighter brown color of the spent biomass remaining after UAE as compared to the one left after SLE_(II)_ (Figure 5).

The cumulative increase in phenolic compound extraction observed during the cascade PEF-UAE process, as depicted in Figure 4, can be attributed to the electroporation and cavitation effects induced by PEF and US treatments, respectively. These effects have the potential to further augment the degree of cell disintegration in cherry pomace tissues, thereby progressively enhancing the leakage of intracellular compounds [26].

However, it is noteworthy that most of the phenolic compounds were retrieved following the first PEF-assisted extraction step (Figure 4a–c). It appears that relatively small molecules such as phenolic compounds (with molecular weights ranging from 0.5 to 4 kDa) can readily traverse the pores created by PEF treatment during the solvent extraction, leading to a high recovery yield.

This is supported by previous studies demonstrating that PEF efficiently unlocks carbohydrates and low-molecular-weight proteins (<20 kDa) from microalgae while being less efficient in extracting larger molecular-weight proteins (20–200 kDa), which require higher-intensity cell disruption methods such as HPH and bead milling [50].

Hence, it can be hypothesized that the pores formed on the cell membrane of cherry pomace tissues during PEF treatment are large enough to facilitate the release of a significant amount of phenolic compounds. Subsequently, the additional application of US treatment allows for the release of additional phenolic compounds that may have been trapped inside the cell or bound to the cell wall [50]. However, further investigation is strongly recommended to confirm this hypothesis.

The findings presented here are in line with the research conducted by Grimi et al. [51], who explored the utilization of PEF in a sequential cascade mode alongside other disruption techniques like high-voltage electrical discharges (HVED), US, and HPH to enhance the extraction of proteins from microalgae. Grimi et al. found that HPH was the most efficient disruption method, resulting in the highest extraction of proteins (91%), whereas PEF demonstrated an efficiency of 5%, a value surpassing the additional contributions of HVED and US.

Regarding the antioxidant activity of the extracts, the FRAP values of the cascade treatment (Figure 4d) significantly increased with diffusion time, following a similar trend as the one observed for phenolic compounds (Figure 4a–c), regardless of the application of US pre-treatment. However, at each diffusion time, applying US pre-treatment to the residual biomass significantly (*p* ≤ 0.05) increased the FRAP values compared to conventional extraction (SLE_(II)_), indicating enhanced antioxidant potential. The maximum FRAP value was detected after 60 min of extraction for both SLE_(II)_ (153.3 ± 0.12 mgAAE/gDW) and UAE (160.0 ± 0.13 mgAAE/gDW). 

Additionally, a robust positive correlation was noted between the phenolic compounds and FRAP values, with Pearson correlation coefficients ranging from 0.97 to 0.99 for TPC, 0.94 to 0.99 for FC, and 0.89 to 0.95 for TAC. This suggests that the phenolic compounds recovered through the cascade process largely contribute to the overall antioxidant power of the cherry pomace extracts, which is consistent with observations from the extracts of the first step.

The beneficial effects of the PEF and US pre-treatments, applied alone and comparatively, on the extraction of phenolic compounds from various plant matrices such as grape pomace, pomegranate, plum and grape peels, oregano, and wild thyme, have been demonstrated in previous studies [33,46,52,53]. However, few studies have investigated the cascade combination of PEF and US pre-treatments for extracting valuable compounds from plant tissues, and none have specifically explored cherry pomace. Nevertheless, Tzima et al. [53] explored the application of optimized PEF pre-treatment to improve the recovery of phenolics from fresh rosemary and thyme byproducts in a subsequent US-assisted extraction step. They observed enhanced recovery of the phenolic compounds and antioxidant capacity as compared to individual US treatment. Similarly, Manzoor et al. [38] investigated the effect of pulsed electric field (PEF) and ultrasound (US) on almond extracts by combining these technologies sequentially, which resulted in higher values for TPC, FC, condensed tannins, TAC, and antioxidant activity as compared to single PEF or US treatment and conventional extraction of untreated samples.

Based on the achieved results, further experiments and HPLC-DAD analysis were carried out to quantify the major phenolic compounds recovered after 60 min of extraction from both the individual and cascade extraction processes.

### 3.3. Quantification of the Main Phenolic Compounds via HPLC-DAD Analysis

The phenolic composition of the cherry pomace extracts obtained in a 50% (*v*/*v*) ethanol-water mixture following 60 min of extraction at 50 °C was evaluated using HPLC-DAD analysis. The samples included extracts achieved after individual SLE_(I)_ and PEF-assisted extraction processes, as well as those obtained through a cascade approach (PEF+SLE_(II)_ and PEF+UAE).

Chromatogram profiles illustrating the results are presented in Figure 6 and Figure 7, with the concentrations of identified phenolic compounds detailed in Table 1.

Overall, the results showed that, regardless of whether the PEF and US pre-treatments were applied, the extracts primarily comprised three phenolic compounds: neochlorogenic acid, cyanidin-3-*O*-rutinoside, and rutin. Additionally, smaller quantities of 4-*p*-coumaroylquinic acid, 3,5-dicaffeoylquinic acid, chlorogenic acid, and cyanidin-3-*O*-glucoside were detected (Table 1).

These findings align with prior research indicating that sweet cherries and their pomace are abundant in hydroxycinnamic acid derivatives, notably chlorogenic acid derivatives, and 4-*p*-coumaroylquinic acid, which are the primary colorless polyphenols found in sweet cherry fruit [12,41,54]. Additionally, they contain quercetin derivatives such as rutin [12,55]. Major anthocyanins found in sweet cherries include cyanidin-3-*O*-rutinoside (resulting in a red-purple color) and cyanidin-3-*O*-glucoside (producing an orange-red color), followed by peonidin-3-*O*-rutinoside (also giving an orange-red color) [12,55].

Notably, according to the results presented in Figure 6, mild PEF pre-treatment conditions did not alter the type and quantity of the phenolic compounds detected in the extracts. This consistency aligns with observations from various studies on different red fruit residues, including cherry pomace. Researchers found that the extracts from untreated and PEF-treated samples exhibited similar phenolic profiles, indicating that no degradation occurred with PEF pre-treatments [12,30,33,34,48,56,57]. However, further investigations and analysis are warranted to confirm the absence of any formation of undesired compounds following the application of PEF treatment.

It is noteworthy that PEF treatment significantly increased the amounts of all individual phenolic compounds as compared to the untreated samples (Table 1). This increment, however, occurred in a manner dependent upon the specific phenolic compounds, as it was more evident for rutin (272%), 3,5-dicaffeoylquinic acid (206%), chlorogenic acid (164%), and cyanidin-3-*O*-glucoside (160%) than for 4-*p*-coumaroylquinic acid (91%,) cyanidin-3-*O*-rutinoside (81%), and neochlorogenic acid (51%) (Table 1). As reported by Pataro et al. [47], this phenomenon could be explained by the selective nature of PEF pre-treatment, which induces the extraction of certain compounds based on factors like the specific group of polyphenols, their distribution within the plant tissues, and their affinity to the plant matrix [56,58]. This phenomenon underscores the importance of considering selective extraction methods for obtaining extracts with specific biochemical profiles.

Figure 7 displays the chromatogram profiles of the phenolic compounds extracted in the second step by SLE_(II)_ and UAE from the residual cherry pomace remaining after the PEF-assisted extraction. As it can be inferred, the integration of UAE into the cascade scheme (PEF+UAE) resulted in the further release of phenolic compounds, which significantly enhanced the detected amounts of chlorogenic acid (240%), 4-p-coumaroylquinic acid (220%), cyanidin-3-*O*-glucoside (175%), cyanidin-3-*O*-rutinoside (84%), rutin (76%), neochlorogenic acid (75%), and 3,5-dicaffeoylquinic acid (75%) as compared to conventional extraction (PEF+SLE_(II)_), without causing degradation phenomena (Figure 7, Table 1).

These findings are consistent with those reported by Kaur et al. [59], demonstrating that the use of US technology on java plum pomace significantly boosts the extraction yield of various phenolic compounds, including gallic acid, 4-amino benzoic acid, catechin hydrate, 3-hydroxybenzoic acid, vanillic acid, 2,3-dihydroxybenzoic acid, *p*-coumaric acid, quercetin, and kaempferol. Similarly, Carpentieri et al. [22] observed increased concentrations of carvacrol and thymol in oregano and thyme extracts following US pre-treatment as compared to control samples.

However, it is worth noting that the proportions of different phenolic compounds extracted after individual PEF and cascade PEF-UAE treatments were influenced by the cell disintegration technique employed. This discrepancy may be partly explained by the differing mechanisms of action of PEF and US. The cavitation phenomena of US technology could facilitate the release of phenolic compounds bound to other matrix components in complex forms, which PEF pre-treatment may not achieve. This observation is somehow consistent with previous research by Carpentieri et al. [22], where GC/MS analysis of oregano and thyme extracts obtained after individual PEF and US-assisted extraction processes yielded similar chromatogram profiles of phenolic compounds, albeit with varying proportions depending on the cell disintegration technique used, which indicated differing selectivity.

The findings from the HPLC-DAD analysis validate the results depicted in Figure 2, Figure 3 and Figure 4, affirming the efficacy of the cascade approach in enhancing the extraction of valuable phenolic compounds from the intracellular space of cherry pomace tissues. It was observed that the majority of phenolic compounds were uncloaked during the initial extraction stage utilizing individual PEF pre-treatment, with subsequent UAE steps facilitating the recovery of additional amounts of the same phenolic compounds that remained trapped within plant cells.

### 3.4. Impacts of PEF and US Pre-Treatment on the Microstructure of Cherry Pomace Tissue

In this study, scanning electron microscopy (SEM) was utilized to investigate the sequential effects of pre-treatments involving PEF and US on the microstructures of cherry pomace tissues and the extractability of phenolic compounds.

Figure 8 illustrates SEM images depicting cherry pomace tissues following the industrial pressing of the fruits (control), followed by various extraction methods: conventional solvent liquid extraction (SLE_(I)_), PEF-assisted extraction alone (c), and cascade treatments combining PEF with SLE_(II)_ and PEF with UAE.

Significant differences were observed in the microstructures of treated versus untreated samples. The industrial cherry pomace (Figure 8a) displayed a rippled surface that likely stemmed from the mechanical damages incurred during fruit pressing. Despite this, it retained some smoothness, presenting a visually compact appearance that could hamper solvent penetration and intracellular compound transfer, consistent with previous findings (Figure 2 and Figure 3).

After the first conventional extraction step (SLE_(I)_), the residual cherry pomace surfaces (Figure 8b) exhibited slight rippling and wrinkling as compared to fresh pomace, retaining some compactness. These changes likely resulted from the action of the ethanol-based solvent and the moderate extraction temperature, which could affect the barrier properties of the cell membrane and promote intracellular matter leakage [22,30].

The PEF-assisted SLE led to notable microstructural alterations (Figure 8c), including cell shrinkage, increased surface roughness, and the formation of depressions (cavity) and microchannels, indicating a more porous structure. These changes likely resulted from a cellular response to PEF treatment and subsequent intracellular compound leakage, which, combined with solvent exposure, contributed to increased phenolic compound extraction, which is consistent with previous studies. 

These effects likely stem from the cellular response to the PEF pre-treatment and the subsequent leakage of a significant amount of intracellular compounds through the electroporated cell membranes, which, in some cases, resulted in cell collapse. Moreover, the exposure of the PEF-treated cherry pomace to the ethanol solvent and moderate temperatures during the subsequent 60 min extraction may have further contributed to the observed changes. These observations are supported by the high extraction yield of phenolic compounds detected in the extracts from the PEF-treated samples compared to the control extraction, as illustrated in Figure 2 and Figure 3. Furthermore, these findings align with those of Li et al. [60], who noted that PEF treatment induced significant changes in the cell structure of mushrooms, including the development of intracellular space and larger cavities, enhancing the mass transfer of water during subsequent drying and rehydration processes. Similarly, Carullo et al. [24] observed shrinkage of PEF-treated microalgae cells, attributed to pore formation in the cell membranes and the consequent leakage of intracellular compounds. 

In the case of cascaded treatments, applying the SLE process in the second phase (designated as PEF+SLE_(II)_) to the residual biomass from the PEF-treated cherry pomace resulted in changes similar to those seen in the individual PEF-assisted extraction step (Figure 8d). However, it is noteworthy that the SLE_(II)_ process slightly heightened irregularities on the cell surface, leading to even rougher surfaces and more pronounced internal depressions and cavities within the plant cells. It is plausible that the continued action of organic solvents and moderate temperatures during this additional SLE step contributed to the erosion and weakening of the cell wall/membrane system and facilitated diffusion processes, thereby impacting the efficacy of the second extraction step (Figure 4). On the other hand, the additional application of UAE to the PEF-treated residual biomass (designated as (PEF+UAE) induced significant additional changes in the structure of cherry pomace. This resulted in a less compact and more porous structure compared to the individual SLE_(I)_, PEF, and combined PEF-SLE_(II)_ treatments, showing the presence of larger cavities (Figure 6e). These effects can be attributed to the collapse of cavitation bubbles, which may induce a combination of phenomena such as fragmentation, localized erosion, pore formation, shear force, increased absorption, and swelling index in the cellular matrix of the plant. This facilitates solvent-tissue contact and intracellular compound release, as indicated by the results reported in Figure 4 and Figure 5.

These findings are consistent with a study by Li et al. [60], which reported that the combined pre-treatment (PEF-US) on mushrooms exhibited a higher number and larger pore size as compared to individual PEF and US applications, thereby increasing the drying rate. Similarly, Seremet et al. [61] reported a higher number of pores in dried carrot tissue with the combined application (PEF-US).

The results support the crucial role played by the cascaded application of PEF and US pre-treatments in enhancing the extractability of valuable compounds from industrial cherry pomace, thus contributing to its potential valorization.

## 4. Conclusions

The results obtained in this study reveal the efficacy of employing a two-stage cascaded scheme involving PEF and US pre-treatments for enhancing and intensifying the extractability of high-value-added compounds from industrial cherry pomace. 

Notably, the PEF pre-treatment employed during the initial extraction phase played a crucial role in permeabilizing cell membranes. This facilitated the extraction of approximately 80% of the total phenolic compounds recovered from cherry pomace following the PEF-UAE cascade process. Consequently, there was a remarkable increase of 293% for TPC, 304% for FC, 380% TAC, and 274% for FRAP when compared to untreated samples (SLE_(I)_). This highlights the potential of PEF-assisted extraction in enhancing the extractability of most phenolic compounds contained in the intracellular space of cherry pomace tissues without the need for hazardous organic solvents.

The subsequent implementation of UAE in the second extraction step resulted in an additional 20% increase in extraction yield. This was particularly notable with increases of 21% for TPC, 49% for FC, 56% for TAC, and consequently, an enhancement in antioxidant power (FRAP) by 26% compared to untreated samples (SLE_(II)_). These findings underscore the capability of US technology to further disrupt electropermeabilized cell tissues, facilitating the release of additional intracellular compounds that PEF alone could not unlock.

HPLC-DAD analyses confirmed that the cascade PEF-UAE process significantly improved the extraction yields of various phenolic compounds, with neochlorogenic acid, cyanidin-3-*O*-rutinoside, and rutin being predominant. Importantly, no selective extraction of specific phenolic compounds was observed under the investigated processing conditions.

Scanning Electron Microscopy (SEM) revealed progressive microstructural changes induced by the sequential application of PEF and US pre-treatment, resulting in increased porosity, larger cavities, and microchannels, which correlated with the improved extractability of phenolic compounds exhibited by the cascade PEF-UAE process.

This proposed cascade approach not only enhances economic profitability by maximizing the use of industrial cherry pomace but also aligns with environmental sustainability goals by utilizing green solvents, minimizing waste, and fully valorizing biomass resources. However, successful implementation of this approach necessitates addressing challenges related to technological complexities, upscaling, and additional costs associated with innovative technologies. Furthermore, exploring the potential extension of this cascade approach to other plant-based biomasses warrants further investigation.

## Figures and Tables

**Figure 1 foods-13-01043-f001:**
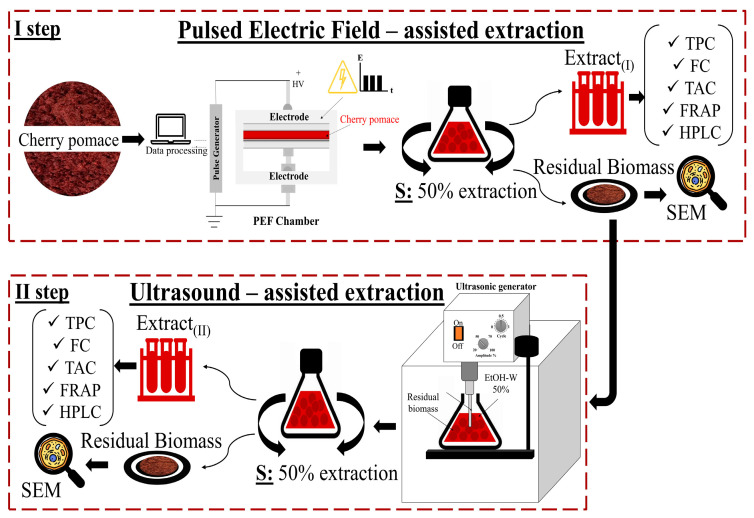
Schematic representation of the “cascade approach” of cherry pomace proposed in this study. First extraction step: PEF-assisted extraction; second extraction step: UAE. EtOH-W: ethanol-water mixture.

**Figure 2 foods-13-01043-f002:**
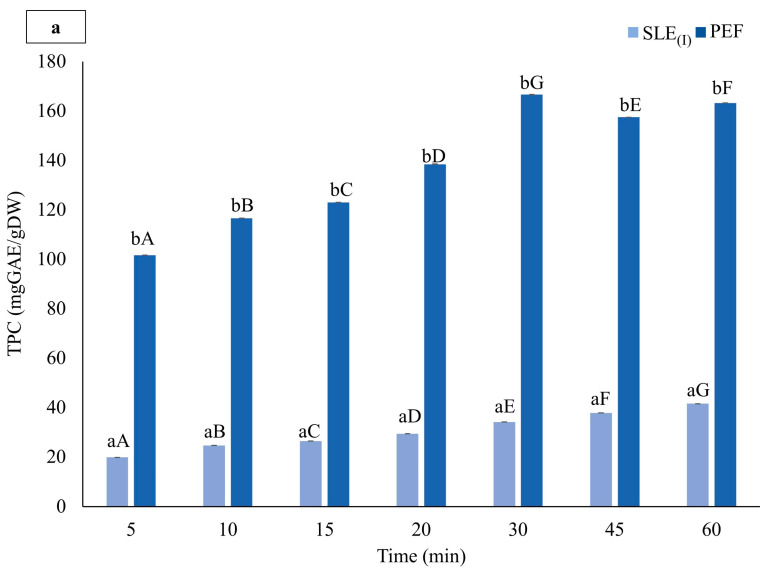
(**a**) Total phenolic content (TPC), (**b**) flavonoid content (FC), (**c**) total anthocyanins content (TAC), and (**d**) antioxidant activity (FRAP) of extracts from untreated (SLE_(I)_) and PEF-treated (*E* = 3 kV/cm, *W_T_* = 10 kJ/kg) cherry pomace in a 50% (*v*/*v*) ethanol-water mixture as a function of the extraction time. The extraction temperature was set at 50 °C. The data are presented as mean ± standard deviation (*n* = 9). Lowercase letters on the bars denote significant differences (*p* ≤ 0.05) between extracts from untreated and PEF-treated cherry pomace at the same extraction time, while uppercase letters denote significant differences (*p* ≤ 0.05) among extracts from either untreated or PEF-treated cherry pomace at different extraction times.

**Figure 3 foods-13-01043-f003:**
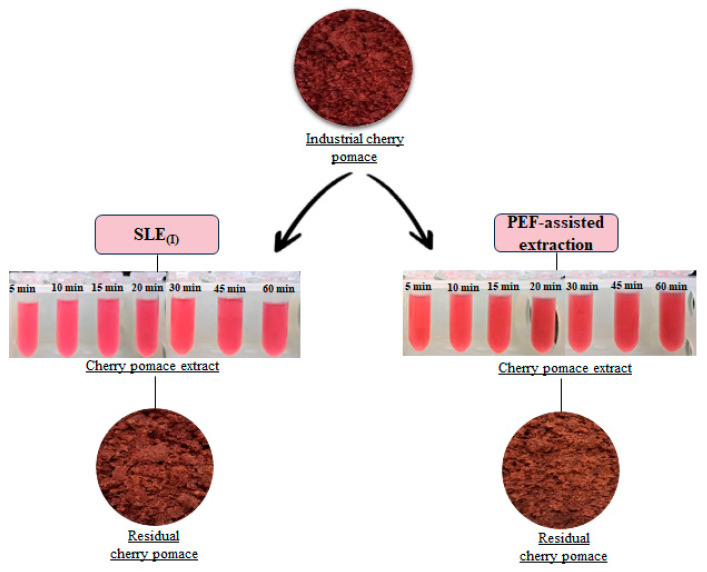
Photographs of industrial cherry pomace, 50% (*v*/*v*) ethanol water extracts obtained after the SLE_(I)_ and PEF-assisted extraction process, and corresponding residual cherry pomace.

**Figure 4 foods-13-01043-f004:**
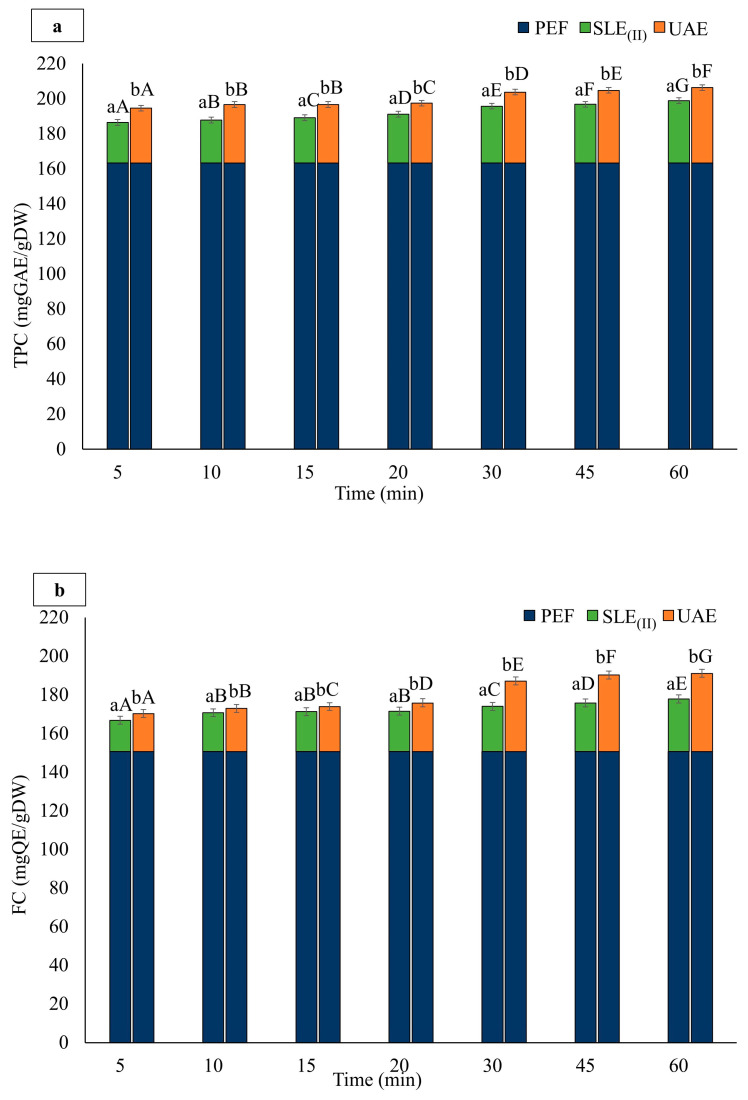
Cascade process: Cumulative (**a**) total phenolic content (TPC), (**b**) flavonoid content (FC), (**c**) total anthocyanins content (TAC), and (**d**) antioxidant activity (FRAP) of the extracts, achieved after the PEF (*E* = 3 kV/cm, *W_T_* = 10 kJ/kg) treatment of cherry pomace followed by either solid-liquid extraction (SLE_(II)_) or US (*P* = 200 W, *t* = 20 min)-assisted extraction (UAE) of residual cherry pomace in 50% (*v*/*v*) ethanol-water mixture as a function of the extraction time. Other extraction conditions are as follows: T = 50 °C, pH = 2.5, and S/L = 1/5 g/mL. The data are presented as mean ± standard deviation (*n* = 9). Bars labeled with different lowercase letters represent significant differences (*p* ≤ 0.05) between the extracts from cherry pomace treated with PEF+SLE_(II)_ and PEF+UAE at the same extraction time. Bars labeled with different uppercase letters represent significant differences (*p* ≤ 0.05) among the extracts from either PEF+SLE_(II)_- or PEF+UAE-treated cherry pomace at different extraction times.

**Figure 5 foods-13-01043-f005:**
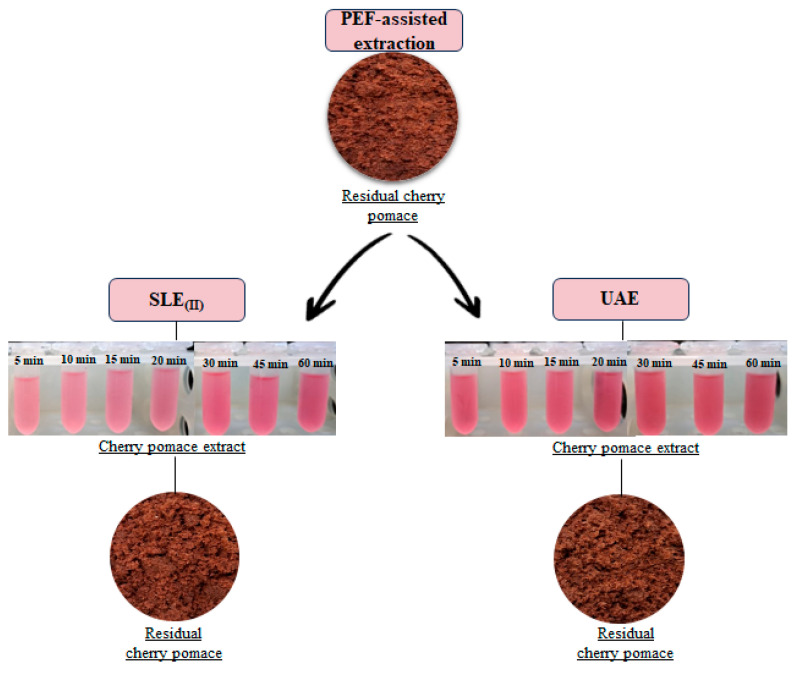
Photographs of residual cherry pomace left after the PEF-assisted extraction process, 50% (*v*/*v*) ethanol water extracts obtained after SLE_(II)_ and UAE, and corresponding residual cherry pomace.

**Figure 6 foods-13-01043-f006:**
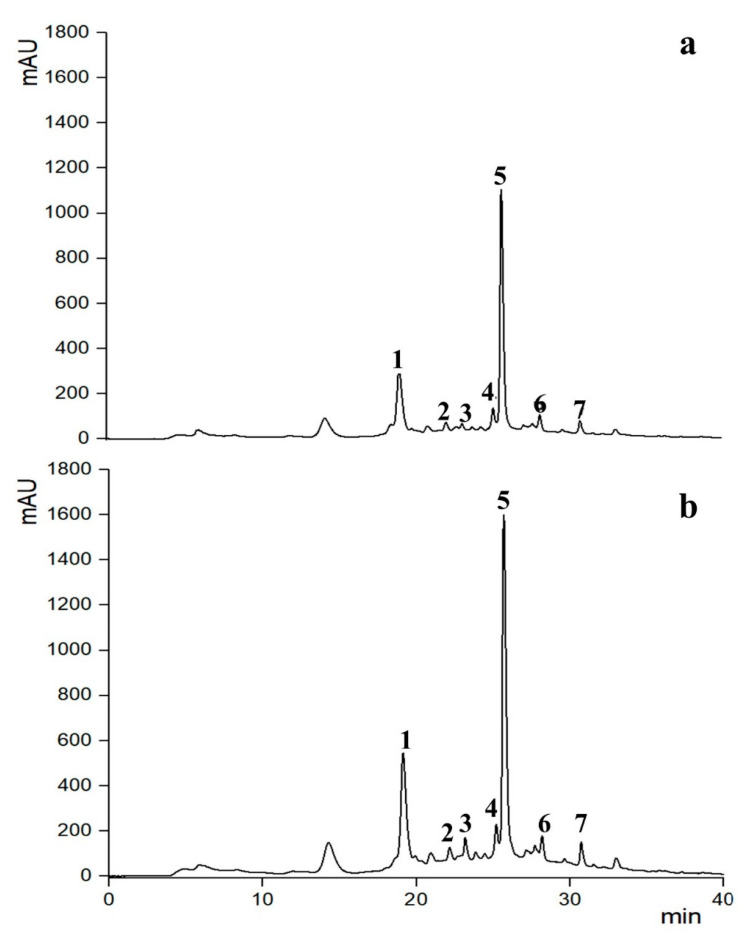
First extraction step: HPLC-DAD chromatograms (λ = 280 nm) of 50% (*v*/*v*) EtOH-W extracts obtained after 60 min of extraction at 50 °C from (**a**) untreated (SLE(I)) and (**b**) PEF-treated (E = 3 kV/cm, W_T_ = 10 kJ/kg) cherry pomace. Peak identification: (1) neochlorogenic acid, (2) chlorogenic acid, (3) 4-*p*-coumaroylquinic acid, (4) cyanidin-3-*O*-glucoside, (5) cyanidin-3-*O*-rutinoside, (6) 3,5-dicaffeoylquinic acid, and (7) rutin.

**Figure 7 foods-13-01043-f007:**
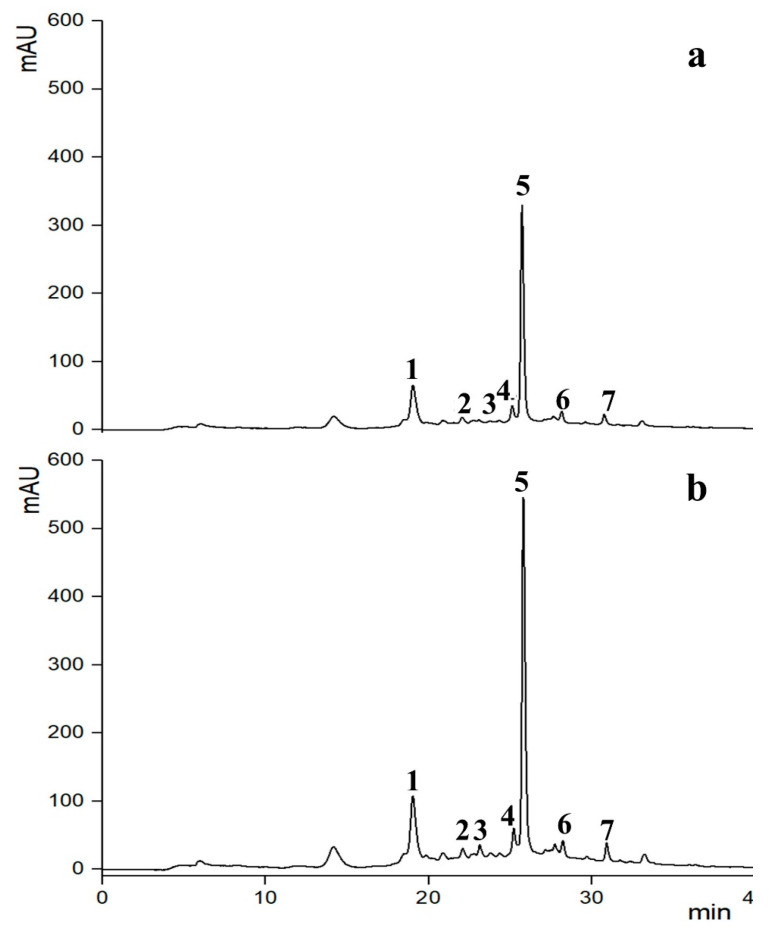
Second extraction step: HPLC-DAD chromatograms (λ = 280 nm) of 50% (*v*/*v*) EtOH-W extracts obtained after 60 min of extraction at 50 °C from (**a**) solid-liquid extraction (SLE_(II)_) and (**b**) US (*P* = 200 W, *t* = 20 min)-assisted extraction (UAE) of residual cherry pomace left after PEF (*E* = 3 kV/cm, *W_T_* = 10 kJ/kg)-assisted extraction process. Peak identification: (1) neochlorogenic acid, (2) chlorogenic acid, (3) 4-*p*-coumaroylquinic acid, (4) cyanidin-3-*O*-glucoside, (5) cyanidin-3-*O*-rutinoside, (6) 3,5-dicaffeoylquinic acid, and (7) rutin.

**Figure 8 foods-13-01043-f008:**
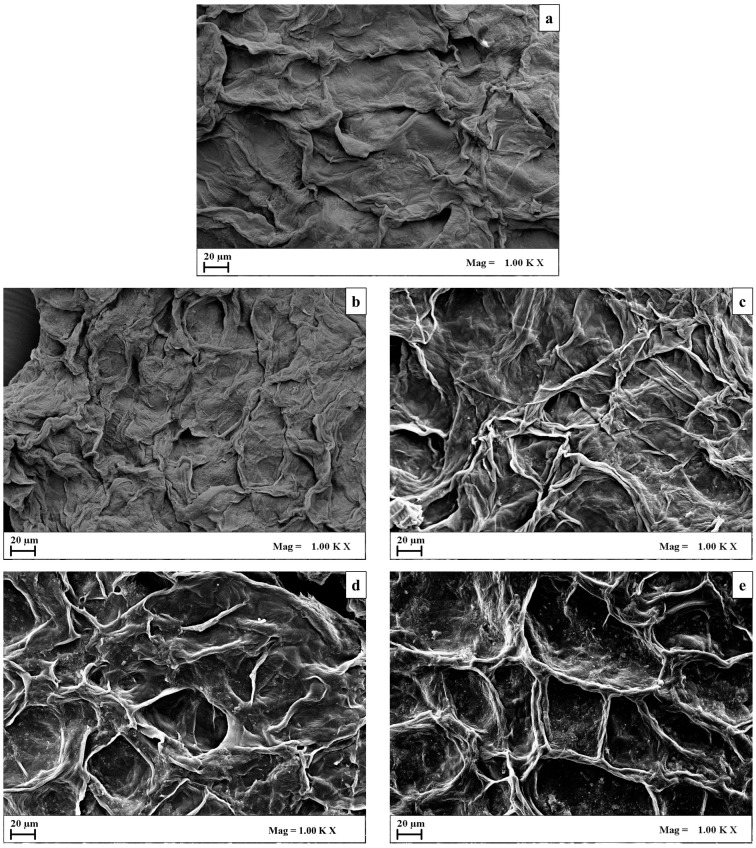
Scanning electron microscopy (SEM) of cherry pomace tissues obtained after industrial pressing of the fruits (**a**) and after the application of SLE_(I)_ alone (**b**), PEF-assisted extraction alone (**c**), and combined PEF+SLE_(II)_ (**d**), and PEF+UAE (**e**).

**Table 1 foods-13-01043-t001:** Concentrations (mg/g_FW_) of neochlorogenic acid, chlorogenic acid, *4-p*-coumaroylquinic acid, cyanidin-*3-O*-glucoside, cyanidin-*3-O*-rutinoside, 3,5-dicaffeoylquinic acid, and rutin detected (HPLC-DAD analysis) in the extracts obtained (I) after 60 min of extraction at 50 °C from untreated (SLE_(I)_) and PEF-treated (*E* = 3 kV/cm, *W_T_* = 10 kJ/kg) cherry pomace and (II) after 60 min of extraction at 50 °C from solid-liquid extraction (SLE_(II)_) and US (*P* = 200 W, *t* = 20 min, *f* = 24 kHz, *T* = 25 °C ± 1 °C)-assisted extraction (UAE) of residual cherry pomace left after PEF (*E* = 3 kV/cm, *W_T_* = 10 kJ/kg)-assisted extraction process.

Extraction Step	Extraction Method	Neochlorogenic Acid (mg/g_DW_)	Chlorogenic Acid (mg/g_DW_)	4-*p*-coumaroylquinic Acid (mg/g_DW_)	Cyanidin-3-*O*-glucoside (mg/g_DW_)	Cyanidin-3-*O*-rutinoside (mg/g_DW_)	3,5-Dicaffeoylquinic Acid (mg/g_DW_)	Rutin (mg/g_DW_)
**I**	SLE_(I)_	14.60 ± 0.61	1.44 ± 0.17	2.52 ± 0.22	0.80 ± 0.03	7.84 ± 0.41	1.96 ± 0.03	2.72 ± 0.21
	PEF	22.00 ± 1.12	3.8 ± 0.22	4.80 ± 0.27	2.08 ± 0.12	14.16 ± 0.54	6.00 ± 0.51	10.12 ± 0.65
**II**	SLE_(II)_	2.04 ± 0.23	0.20 ± 0.6	0.20 ± 0.04	0.16 ± 0.02	1.92 ± 0.27	0.32 ± 0.02	0.68 ± 0.12
	UAE	3.56 ± 0.35	0.68 ± 0.03	0.64 ± 0.08	0.44 ± 0.04	3.48 ± 0.90	0.56 ± 0.06	1.16 ± 0.04
**Total**	PEF+SLE_(II)_	24.04 ± 0.98	4.00 ± 0.14	5.00 ± 0.41	2.24 ± 0.09	16.08 ± 1.13	6.32 ± 0.83	10.8 ± 0.66
	PEF+UAE	25.56 ± 1.21	4.48 ± 0.09	5.44 ± 32	2.52 ± 0.11	17.64 ± 0.98	6.56 ± 0.74	11.28 ± 0.89

## Data Availability

The original contributions presented in the study are included in the article/Appendix A, further inquiries can be directed to the corresponding author.

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
