# Peer review of "Sustainable Valorization of Industrial Cherry Pomace: A Novel Cascade Approach Using Pulsed Electric Fields and Ultrasound Assisted-Extraction"

_foods, 2024, doi:10.3390/foods13071043_

Round 1
Reviewer 1 Report
Comments and Suggestions for Authors
Dear Authors,
The paper I read offered insightful knowledge on the current trend of valorizing industrial cherry pomace, a topic that's generating significant interest. However, there are a few comments that should be considered.
The abstract is too long, it should be reduced to 250 words. Also, the number of keywords should be reduced to 5. In line 24., It can be interpreted that you are referring to SLE (I), instead of SLE (II), try to emphasize that you are referring to SLE (II). The Latin name of sweet cherry should be formatted in italics. In line 149., Extract(I) has been mentioned only here and nowhere else throughout the text. Either discard this denomination or use Extract(I) when referring to the extracts obtained after PEF pre-treatment. It would also add to clarity if you would write in the next sentence that PEF pre-treatment + SLE, will later on be referred to as PER-assisted extraction. It is important to state, in line 153., that the used residual biomass that underwent the second step, was the biomass after PEF-assisted extraction which yielded the highest bioactive content. In line 265., since SLE(I) refers to the type of extraction, it should be left out because of the ending of the sentence. In line 286., to state that certain conditions are optimal for the extraction process, it is necessary to include a more detailed analysis which includes measurements of energy consumption, time, and yield of the extraction. In my opinion it is better to avoid saying “optimal” in this case. In the case of SLE(I), TAC was declining with the prolonged extraction time. Since FC in the case of SLE(I) and PEF-extraction is aligned, what could cause a discrepancy in the TAC? Provide possible explanation. In line 325., exclude “somehow” and put more accurate word. In line 381., change units from subscript to normal text. In line 391., exclude the last part of the sentence after comma. The sentence in lines 505-508 should be excluded from the text. The sentence in lines 514-520 is repetition of previous one. In line 525., sort phenolic compounds in decreasing order. In line 648., percentages compared to SLE(II). It would also be meaningful to compare, TPC, FC, TAC, and FRAP at the end of the process to the SLE(I), and involve those results here and in section 3.2. The number of references should be reduced to 60.
Kind regards
Comments on the Quality of English Language
Minor editing of English language required.
Author Response
Reviewer #1
Comments to the Author
Dear Authors,
The paper I read offered insightful knowledge on the current trend of valorizing industrial cherry pomace, a topic that's generating significant interest. However, there are a few comments that should be considered.
- The abstract is too long, it should be reduced to 250 words.
Answer: The abstract has been revised accordingly, reducing it to less than 250 words without compromising the clarity and completeness of the information.
- Also, the number of keywords should be reduced to 5.
Answer: The number of keywords has been reduced to five in the revised version.
- In line 24., It can be interpreted that you are referring to SLE (I), instead of SLE (II), try to emphasize that you are referring to SLE (II).
Answer: Now, the sentence has been revised to emphasize that we are specifically referring to the second stage of conventional solid-liquid extraction (SLE).
- The Latin name of sweet cherry should be formatted in italics.
Answer: The Latin name of the sweet cherry has been properly formatted in italics.
- In line 149., Extract(I) has been mentioned only here and nowhere else throughout the text. Either discard this denomination or use Extract(I) when referring to the extracts obtained after PEF pre-treatment. It would also add to clarity if you would write in the next sentence that PEF pre-treatment + SLE, will later on be referred to as PER-assisted extraction.
Answer: We agree with the Reviewer’s comment. However, to maintain a certain clarity in Figure 1, we decided to maintain the denomination Extract(I) and Extract (II), but we specified that the first and second output stream were labelled as Extract(I) and Extract(II) in Figure 1.
Furthermore, we have already clarified in the manuscript that the combination of PEF pre-treatment and SLE will be referred to as "PEF-assisted extraction" throughout the text.
- It is important to state, in line 153., that the used residual biomass that underwent the second step, was the biomass after PEF-assisted extraction which yielded the highest bioactive content.
Answer: In this case, we kept the previous text as it already explicitly state that the residual biomass left after PEF-assisted extraction underwent the second extraction step. However, according to the Reviewer’s comment, we added an explanation that this second extraction step was carried out with the aim of further enhancing the recovery yield of bioactive compounds..
- In line 265., since SLE(I) refers to the type of extraction, it should be left out because of the ending of the sentence.
Answer: The word SLE(I) has been removed accordingly.
- In line 286., to state that certain conditions are optimal for the extraction process, it is necessary to include a more detailed analysis which includes measurements of energy consumption, time, and yield of the extraction. In my opinion it is better to avoid saying “optimal” in this case.
Answer: We agree with the Reviewer. The word "optimal" has been removed accordingly.
- In the case of SLE(I), TAC was declining with the prolonged extraction time. Since FC in the case of SLE(I) and PEF-extraction is aligned, what could cause a discrepancy in the TAC? Provide possible explanation.
Answer: If we look at the results of Figure 2, in addition to what observed by the reviewer, it can be seen that at the higher extraction time, also results of TPC for SLE and PEF samples show a slightly different trends. Sincerely it is not ease to explain this different behavior, which could depend on the complex interaction between the extracted compounds, solvent and processing conditions. In our opinion, further studies should be carried out specifically devote to investigate the effect of extraction conditions (with and without the application of PEF) on the degradation of target compounds.
In line 325., exclude “somehow” and put more accurate word.
Answer: The term 'somehow' has been removed.
- In line 381., change units from subscript to normal text.
Answer: The units have been changed from subscript to normal text.
- In line 391., exclude the last part of the sentence after comma.
Answer: The required change has been made accordingly.
- The sentence in lines 505-508 should be excluded from the text.
Answer: The sentence in lines 505-508 has been removed accordingly.
- The sentence in lines 514-520 is repetition of previous one.
Answer: The sentence in lines 514-520 has been deleted.
- In line 525., sort phenolic compounds in decreasing order.
Answer: The phenolic compounds have now been sorted in decreasing order.
- In line 648., percentages compared to SLE(II). It would also be meaningful to compare, TPC, FC, TAC, and FRAP at the end of the process to the SLE(I), and involve those results here and in section 3.2.
Answer: According to this comment, the text has been revised. We have now included in the Conclusion section the increment in the extraction yields of total phenolic content (TPC), flavonoid content (FC), total anthocyanin content (TAC), and antioxidant activity (FRAP) induced by the pulsed electric field (PEF) pre-treatment compared to the SLE(I). This information was initially presented in section 3.1.
The number of references should be reduced to 60.
The number of references has been reduced to 61. The authors believe that the remaining citations are essential and should not be removed.
Reviewer 2 Report
Comments and Suggestions for Authors
Ms: Foods-2892939
Dear editor,
The manuscript presents useful information about the possible valorization of cherry pomace by pulsed electric fields and ultrasounds, but some considerations should be solved.
- Introduction. Lines 79-80. “This technique, when optimized, ensures efficient extraction without overheating the sample. It is not totally true. One of the problems of the application of US is that the technique provokes that the sample to increase its temperature. To avoid it, the authors employ an ice-water bath (line 156). The sentence should be modified.
- About the ice-water bath, was the temperature measured?, At which real temperature were the extractions carried out?
- The authors mention that the US treatments were carried out under optimized conditions, with data not shown. In my opinion, the details about how they were fixed are very important and they should be shown in the manuscript.
- Which extracts were analysed by HPLC? 10 mL were injected into the HPLC system, but the authors mention that 2 mL extracts were collected at different times. In my opinion, 10 mL is a very high volume for HPLC analysis, is this volume correct?
- Sections 2.4.1, 2.4.2, 2.4.3, 2.4.4, and 2.4.5. In all of them, the authors mention that “following the methodology reported by Rrucaj et al. [12]”. This is an article from the same authors that the present article is. They are self-cites. The methodologies that the authors use for the different determinations are from different authors with some modifications. They are detailed in the article “Frontiers in Suitable Food Systems, 2022, 6, 8549682”. This work should be mentioned and not that from Rrucaj et al.
- Results. Section 3.1. Optimization of PEF-assisted extraction process of phenolic compounds from industrial cherry pomace. The authors do not optimize the PEF extraction, they optimize the time during the solvent extraction after the PEF procedure…The title should be modified
- Figures 2a-c. The authors mention significant differences according to time, but, according to the figures, it is not easy to see, even in some cases, it seems that there are not significant differences. Perhaps it could be correctly observed if a table with mean values and results from ANOVA study and Tukey´s test was shown.
- Lines 291-297. In my opinion, figure 3 should be eliminated and colorimetric measurements could be added. Colorimetric instruments provide a set of standardized conditions that help assure consistency and repeatability.
- Lines 393-398. The same that is mentioned above. Figure 5 should be eliminated.
- Figure 4. The colors used for SLE and UAE are very similar, they should be modified. The information about PEF should be eliminated from this figure because it does not add relevant information. Again, it is not easy to see the significant differences. Perhaps a table with mean values and the results from Anova would be more useful.
- Lines 494-499. The authors indicate that no degradation during PEF process was observed on the basis of no additional peaks detected in the extracts and on the bibliography, but no experimental study was carried out. In my opinion, it is too speculative. In order to stablish it an experimental study should be carried out. This paragraph should be modified.
- Lines 514-519, the paragraph is repetitive. The same information is shown in paragraph 500-505. It should be deleted or modified.
- Lines 661-662. It should be modified. No degradation studies have been carried out. “no selective extraction”, it is opposite to “As it can be inferred, the integration of UAE into the cascade scheme (PEF+UAE) resulted in further release of phenolic compounds, significantly enhancing the detected amounts of neochlorogenic acid (75%), chlorogenic acid (240%), 4-p-couma-525 roylquinic acid (220%), cyanidin-3-O-glucoside (175%), cyanidin-3-O-rutinoside (84%), 3,5-dicaffeoylquinic acid (75%), and rutin (76%)….lines 523-526
Author Response
Reviewer #2
Dear editor,
The manuscript presents useful information about the possible valorization of cherry pomace by pulsed electric fields and ultrasounds, but some considerations should be solved.
- Introduction. Lines 79-80. “This technique, when optimized, ensures efficient extraction without overheating the sample. It is not totally true. One of the problems of the application of US is that the technique provokes that the sample to increase its temperature. To avoid it, the authors employ an ice-water bath (line 156). The sentence should be modified.
Answer: We totally agree with the Reviewer. In the previous version of the manuscript, we encompassed the side effect of ultrasound (US) treatment, including it within the term "optimized". However, we recognize the merit of directly specifying this aspect. As such, we have amended the sentence accordingly.
- About the ice-water bath, was the temperature measured?, At which real temperature were the extractions carried out?
Answer: The information has been incorporated into the text. Specifically, the temperature of the sample inside the flask, consisting of the solvent and cherry pomace, was monitored throughout the extraction process. The initial temperature was maintained at 20 ± 1 °C, while the maximum temperature increase observed was approximately 5°C.
- The authors mention that the US treatments were carried out under optimized conditions, with data not shown. In my opinion, the details about how they were fixed are very important and they should be shown in the manuscript.
Answer: According to this comment, in the revised version of the manuscript, the experimental data used to optimize ultrasound (US) treatment conditions were added to the supplementary material. They were not included directly in the manuscript due to the high number of figures.
- Which extracts were analysed by HPLC? 10 mL were injected into the HPLC system, but the authors mention that 2 mL extracts were collected at different times. In my opinion, 10 mL is a very high volume for HPLC analysis, is this volume correct?
Answer: The units of the injected volume reported in the previous version were incorrect. The correct units (10 µL) have now been specified.
- Sections 2.4.1, 2.4.2, 2.4.3, 2.4.4, and 2.4.5. In all of them, the authors mention that “following the methodology reported by Rrucaj et al. [12]”. This is an article from the same authors that the present article is. They are self-cites. The methodologies that the authors use for the different determinations are from different authors with some modifications. They are detailed in the article “Frontiers in Suitable Food Systems, 2022, 6, 8549682”. This work should be mentioned and not that from Rrucaj et al.
Answer: In the revised version of the manuscript, the methodology were properly cited according to the Reviewer’s comment.
- Results. Section 3.1. Optimization of PEF-assisted extraction process of phenolic compounds from industrial cherry pomace. The authors do not optimize the PEF extraction, they optimize the time during the solvent extraction after the PEF procedure…The title should be modified
Answer: The Title e thank the Reviewer#2 for the suggestion. The title of the Section 3.1. has now been modified in the revised version of the manuscript. Now it reads: “PEF-assisted extraction process of phenolic compounds from industrial cherry pomace: optimization of the solid-liquid extraction step”.
- Figures 2a-c. The authors mention significant differences according to time, but, according to the figures, it is not easy to see, even in some cases, it seems that there are not significant differences. Perhaps it could be correctly observed if a table with mean values and results from ANOVA study and Tukey´s test was shown.
Answer: We agree with the Reviewer that in some cases, from the plot of Figure 2 could seems that there are not significant differences. However, upon careful reevaluation of the statistical analysis, we reaffirm that the results remain consistent. While some mean values may appear similar, the differences are statistically significant, likely due to the very low standard deviations.
In response to the Reviewer's request, we opted against adding a table containing mean values and results from the ANOVA study, as this would duplicate the data already presented in Figure 2, adding no new information. However, we are available to share the raw data with the Reviewer if necessary.
- Lines 291-297. In my opinion, figure 3 should be eliminated and colorimetric measurements could be added. Colorimetric instruments provide a set of standardized conditions that help assure consistency and repeatability.
Lines 393-398. The same that is mentioned above. Figure 5 should be eliminated.
Answer: We also agree with the Reviewer that colorimetric measurements provide quantitative data that could help to better state the relationship between phenolic content in the extracts and its color intensity.
Unfortunately, samples are no more available and thus we cannot add this data. However, we will consider this suggestion carefully for future work.
Nevertheless, the primary aim of this study is to investigate the sequential and cumulative effects of PEF and US-assisted extraction in a cascade approach, aimed at enhancing the recovery of total phenolic compounds, flavonoids, and anthocyanins from industrial cherry pomace. To further elucidate the extraction process, we have included Figures 3 and 5 in the manuscript for two main purposes: firstly, to provide a schematic representation of the extraction process, and secondly, to visually demonstrate the intensification of color in the extracts from cherry pomace as a function of extraction time and method, along with the visual appearance of the residual biomass. We believe that this additional information can assist readers in better comprehending the cascade approach employed, as well as the influence of time and extraction method on the resultant extracts.
Therefore, we have chosen to retain Figures 3 and 5 in the manuscript for these reasons.
- Figure 4. The colors used for SLE and UAE are very similar, they should be modified. The information about PEF should be eliminated from this figure because it does not add relevant information. Again, it is not easy to see the significant differences. Perhaps a table with mean values and the results from Anova would be more useful.
Answer: The color used for SLE and UAE bars has been adjusted to ensure better distinguishability between them.
We partially agree with the Reviewer regarding the information provided by the PEF data. However, considering that in this paragraph we are presenting result of cascade approach, in our opinion they should be maintained in order to clearly show the contribution of the two extraction step as also described in the text.
Regarding the significant differences, please see the answer to the previous comment on - Figures 2a-c.
- Lines 494-499. The authors indicate that no degradation during PEF process was observed on the basis of no additional peaks detected in the extracts and on the bibliography, but no experimental study was carried out. In my opinion, it is too speculative. In order to stablish it an experimental study should be carried out. This paragraph should be modified.
Answer: With respect to the first part of the Reviwer’s comment, we would specify that, according to literature statement, in the manuscript “no degradation during PEF process” was referred the fact that similar phenolic profiles were detected between untreated and PEF-treated samples rather than no additional peaks were detected in the extracts. However, we agree with the Reviewer that this statement could be speculative. For this reason, a further sentence has been added to specify that additional studies are necessary to confirm the absence of any formation of undesired compounds following the application of PEF treatment.
- Lines 514-519, the paragraph is repetitive. The same information is shown in paragraph 500-505. It should be deleted or modified.
Answer: The paragraph in lines 514-519 has been deleted.
- Lines 661-662. It should be modified. No degradation studies have been carried out. “no selective extraction”, it is opposite to “As it can be inferred, the integration of UAE into the cascade scheme (PEF+UAE) resulted in further release of phenolic compounds, significantly enhancing the detected amounts of neochlorogenic acid (75%), chlorogenic acid (240%), 4-p-couma-525 roylquinic acid (220%), cyanidin-3-O-glucoside (175%), cyanidin-3-O-rutinoside (84%), 3,5-dicaffeoylquinic acid (75%), and rutin (76%)….lines 523-526
Answer: This sentence has been modified accordingly.